# Network Sparsity Unlocks the Scaling Potential of Deep Reinforcement Learning

Guozheng Ma [* 1]   Lu Li [* 2 3]   Zilin Wang [4]   Li Shen [1]   Pierre-Luc Bacon [2 3]   Dacheng Tao [1]

## Abstract

Effectively scaling up deep reinforcement learning models has proven notoriously difficult due to network pathologies during training, motivating various targeted interventions such as periodic reset and architectural advances such as layer normalization. Instead of pursuing more complex modifications, we show that introducing **static network sparsity** alone can unlock further scaling potential beyond their dense counterparts with state-of-the-art architectures. This is achieved through simple one-shot random pruning, where a predetermined percentage of network weights are randomly removed once before training. Our analysis reveals that, in contrast to naively scaling up dense DRL networks, such sparse networks achieve both higher parameter efficiency for network expressivity and stronger resistance to optimization challenges like plasticity loss and gradient interference. We further extend our evaluation to visual and streaming RL scenarios, demonstrating the consistent benefits of network sparsity. Our code is publicly available at GitHub ⭘.

## 1. Introduction

Deep neural networks have demonstrated consistent improvements with increased scale in supervised learning tasks, where larger models reliably yield better results. However, this scaling pattern breaks down in deep reinforcement learning (DRL), where increasing model size often leads to deteriorating performance (Nauman et al., 2024a;b). This limited scalability of DRL models can be largely attributed to severe optimization pathologies that emerge during training (Nikishin, 2024; Nauman et al., 2024a), with these challenges becoming increasingly pronounced as model size grows (Ceron et al., 2024a;b). Specifically, notable pathological behaviors include plasticity loss (Nikishin et al.,

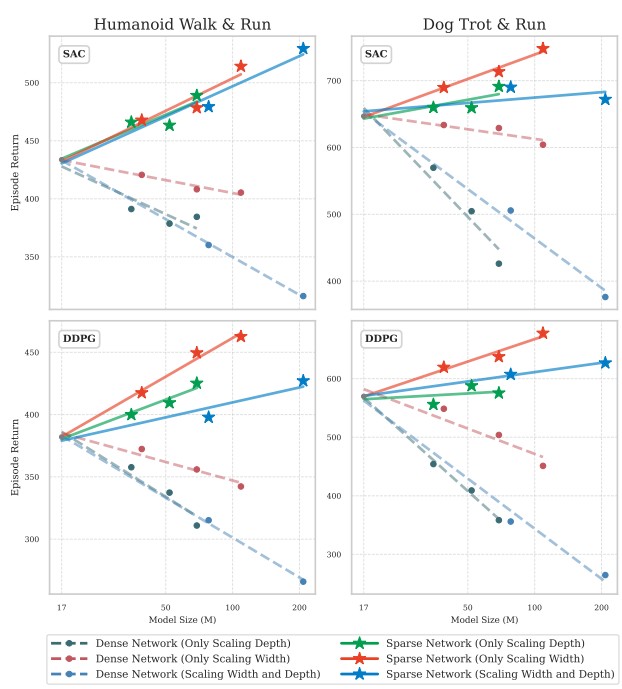

*Figure 1.* Model scaling trends of ★★★**sparse** versus ●●●**dense** networks on four hardest DMC tasks using SimBa architecture with SAC and DDPG. Beyond a ∼17M baseline SimBa network, dense networks (**dashed lines**) exhibit degrading performance with increased scale. In contrast, introducing sparsity while increasing model size (**solid lines**) can unlock further scaling potential.

2022; Sokar et al., 2023), parameter under-utilization (Kumar et al., 2021), capacity collapse (Lyle et al., 2022a), etc.

Recent work has proposed various dynamic approaches to address these pathologies, aiming to break through the scaling barrier of DRL models (Lyle et al., 2022a; Klein et al., 2024). Among these advances, periodic Reset and its variants stand out as a representative approach that enhances model scaling capabilities by mitigating plasticity loss and pathological behaviors through strategic re-initialization of the entire network or specific neurons (Nikishin et al., 2022; Sokar et al., 2023; Xu et al., 2024; Dohare et al., 2024). While effective, these methods require drastic interventions in optimization dynamics, inevitably disrupting training stability and introducing significant computational complexity (Nikishin, 2024; Klein et al., 2024).

Recent architectural advances – particularly spectral normalization (Bjorck et al., 2021), layer normalization (Lyle et al.,

---

[*]Equal contribution  [1]Nanyang Technical University [2]Mila - Quebec AI Institute [3]Université de Montréal [4]University of Oxford. Correspondence to: Li Shen <mathshenli@gmail.com>, Dacheng Tao <dacheng.tao@ntu.edu.sg>.

*Proceedings of the 42nd International Conference on Machine Learning*, Vancouver, Canada. PMLR 267, 2025. Copyright 2025 by the author(s).

2023; 2024a), and residual connections (Espeholt et al., 2018) – have shown considerable success in mitigating plasticity loss and enabling DRL network scaling. Building on these advances, recent works such as BRO (Nauman et al., 2024b) and SimBa (Lee et al., 2024) have achieved significantly improved scalability without requiring Reset operations or RL algorithm modifications. However, while SimBa represents the current state-of-the-art architectural design, its scaling capabilities remain fundamentally limited. Our investigation into scaling SimBa beyond previously studied sizes reveals a consistent pattern: performance deteriorates when scaling the network in any dimension, as evidenced by the sharp drops shown in the dashed lines of Figure 1.

Another line of research aims to leverage adaptive or modified network topologies to enhance the parameter efficiency and scalability of DRL models. In this direction, Ceron et al. (2024a) demonstrates that gradual magnitude pruning of large models leads to dramatic improvements in value-based agents' performance. Ceron et al. (2024b) shows that value-based RL networks equipped with Soft Mixture of Experts (MoEs) (Puigcerver et al., 2024) exhibit improved parameter scalability. In addition, Neuroplastic Expansion (Liu et al., 2024) improves network plasticity by dynamically growing the network from sparse to dense architectures, thereby benefiting from larger model sizes. Although implemented differently, these successful approaches share a crucial insight: introducing network sparsity and topology dynamicity during training holds the potential to enable better parameter scaling in DRL models. However, these studies predominantly propose dynamic methods meant to directly act upon the update steps during optimization while ignoring the static sparsity properties of the network at initialization. Moreover, since existing studies primarily focus on standard MLPs, it remains unclear whether these sparsity benefits extend to modern architectures equipped with residual connections and layer normalization (Lee et al., 2024). Motivated by these current advances in DRL scalability, this paper aims to explore a central question:

> Can static network sparsity alone unlock further DRL model scaling potential beyond current advanced architectures while preventing optimization pathologies?

Through a series of ablation and scaling studies we reveal a clear and definitive answer: **YES!** As summarized in Figure 1, sparse networks continue to show performance gains well beyond the point where their dense counterparts hit scaling limits, demonstrating superior parameter efficiency and enhanced scalability at larger model sizes. Subsequently, Section 4 delves into why introducing sparsity can break through current scaling barriers by leveraging a range of empirical metrics as diagnostic tools. Our analysis reveals that while larger model sizes tend to induce more severe optimization pathologies, appropriate network sparsity ef-

fectively counteracts these negative effects by preventing capacity and plasticity loss (Klein et al., 2024), constraining parameter growth (Lyle et al., 2024b), enhancing simplicity bias (Lee et al., 2024), and mitigating gradient interference (Lyle et al., 2023). Furthermore, in Section 5, we extend our empirical evaluation to visual RL and streaming RL, demonstrating that the benefits of network sparsity consistently generalize across diverse RL setups.

> Contributions of this paper can be summarized as:
>
> 1. While the advanced SimBa architecture (Lee et al., 2024) has greatly improved DRL network scalability, we show that introducing static network sparsity through simple one-shot random pruning (Liu et al., 2022) at initialization can unlock further scaling potential beyond previous limitations.
>
> 2. Our extensive analysis reveals that appropriate network sparsity alone can prevent severe optimization pathologies that emerge as models scale up, such as capacity collapse, plasticity loss, unbounded parameter growth and gradient interference.
>
> 3. We validate that the benefits of network sparsity generalize well across broader RL scenarios.

## 2. Preliminary

In this section, we introduce the development and current best practices in DRL network architecture design aimed at mitigating optimization pathologies. Additionally, as a foundation for our subsequent investigation, we detail our approach to implementing network sparsity. Due to space constraints, a detailed review of pathologies, network scaling, and sparse models in DRL is provided in Appendix A.

### 2.1. Network Architecture Design in Deep RL

Early DRL community primarily treated neural networks as function approximators, focusing research efforts on core RL challenges such as exploration (Ciosek et al., 2019) and value overestimation (Fujimoto et al., 2018) rather than network architecture design. Hence, for a long period, most of DRL works simply employed basic MLPs by default (Fujimoto et al., 2023), adding only a few convolutional layers when processing visual observations (Yarats et al., 2022).

Moreover, the RL paradigm fundamentally differs from (un)supervised learning, with its trial-and-error nature of online interactions and non-stationarity of both data streams and optimization objectives. The interplay of overlooked RL-tailored deep learning mechanisms and inherent RL challenges leads to severe optimization pathologies, which recently have been recognized through several terms, including primacy bias (Nikishin et al., 2022), dormant neuron phenomenon (Sokar et al., 2023), implicit underparameterization (Kumar et al., 2021), capacity loss (Lyle

et al., 2022a), and more broadly, plasticity loss (Klein et al., 2024). More concerning, these pathologies intensify with increasing model scale (Ceron et al., 2024a; Lyle et al., 2023), hindering networks' ability to leverage the enhanced expressivity that larger models should provide, thereby fundamentally limiting the scalability of DRL.

Recent studies have begun to focus on architectural improvements to mitigate these pathologies, progressively pushing forward the effective scaling of DRL networks. Various normalization techniques have shown different degrees of effectiveness, including Spectral Normalization (Bjorck et al., 2021), Batch Normalization (Bhatt et al., 2024), and the widely adopted Layer Normalization (Lee et al., 2023; Lyle et al., 2023; 2024a). Beyond merely using ResNet as a visual encoder (Espeholt et al., 2018), BRO (Nauman et al., 2024b) first demonstrated the effectiveness of incorporating residual blocks in both policy and value networks, significantly enhancing robustness and performance in challenging RL tasks. Drawing from these insights and detailed analysis, SimBa (Lee et al., 2024) further enhanced the training stability by introducing observation normalization layers to regulate input data distributions. Representing the current best practices in DRL architecture design, SimBa successfully scaled DRL models beyond 10M parameters.

However, as shown by Figure 13 in Lee et al. (2024) and our subsequent investigation, further increasing SimBa's model size not only fails to yield performance improvements but also leads to significant degradation. This motivates us to consider: *Have we reached the fundamental scaling limits of DRL models on current benchmarks and tasks?* Or *is there a simple yet effective approach that could push these boundaries further?* Drawing inspiration from previous scalability improvements achieved through non-standard architectures beyond vanilla MLPs that incorporate both network sparsity and dynamicity (Graesser et al., 2022; Tan et al., 2023; Ceron et al., 2024b;a; Liu et al., 2024), this paper decouples sparsity as a standalone feature to examine its effectiveness on top of advanced architectural designs. Our thorough investigation in Section 3 reveals that sparsity alone can unlock further scaling potential. Our empirical study not only establishes a simple yet strong baseline for DRL network scaling, but also suggests untapped opportunities for more specialized DRL architectural designs.

### 2.2. Sparse Network with One-Shot Random Pruning

To isolate the independent role of network sparsity on DRL scaling, we conduct our investigation using static sparse training (SST) (Liu et al., 2022) with one-shot random pruning. This straightforward approach establishes a fixed sparse topology through random pruning before training, avoiding confounding factors such as topology dynamics in dynamic sparse training (DST) (Mocanu et al., 2018; Evci et al., 2020) or targeted topology optimization in pruning at initialization (PaI) (Lee et al., 2019; Hoang et al., 2023).

**Random Pruning.** Static sparse training with one-shot random pruning generates binary masks for each layer at initialization. These masks, which determine the network's sparse topology, remain fixed throughout training. For a network with $L$ layers, each layer $l$ has a binary mask $\mathbf{M}^l \in \{0, 1\}^{n^l \times n^{l-1}}$, where $n^l$ denotes the number of units in layer $l$. The effective weights during both training and inference are computed as $\mathbf{W}^l_{\text{eff}} = \mathbf{M}^l \odot \mathbf{W}^l$, where $\odot$ denotes element-wise multiplication.

**Layer-Wise Sparsity Ratios.** Random pruning represents a remarkably simple random sampling process, requiring only layer-wise sparsity ratios to be pre-defined. There are two commonly adopted approaches for determining layer-wise sparsity ratios from the overall network sparsity:

- *Uniform*: The sparsity ratio $s^l$ of each individual layer $l$ is equal to the overall network sparsity $S$.

- *Erdős-Rényi (ER)*: This approach randomly generates the sparse masks so that the sparsity in each layer $s^l$ scales as $1 - \frac{n^{l-1}+n^l}{n^{l-1}n^l}$ for a fully-connected layer (Mocanu et al., 2018) and as $1 - \frac{n^{l-1}+n^l+w^l+h^l}{n^{l-1}n^lw^lh^l}$ for a convolutional layer with kernel dimensions $w^l \times h^l$ (Evci et al., 2020).

Extensive prior studies in both supervised learning (Liu et al., 2022) and RL (Graesser et al., 2022; Tan et al., 2023; Liu et al., 2024) have shown that ER-based initialization yields superior performance over uniform sparsity, especially at high sparsity levels. Thus, we adopt layer-wise sparsity ratios based on ER throughout our investigation unless specified otherwise.

## 3. Sparsity Promotes DRL Network Scaling

This section aims to investigate whether introducing network sparsity can unlock further scaling potential in DRL models beyond the effective scaling limits of dense SimBa networks. We conducted extensive experiments on several of the most challenging DeepMind Control (DMC) (Tassa et al., 2018) tasks using both Soft Actor-Critic (SAC) (Haarnoja et al., 2018) and Deep Deterministic Policy Gradient (DDPG) (Lillicrap, 2015) with advanced SimBa architecture (Lee et al., 2024). In this section, we first show that appropriate network sparsity both enables further model scaling and enhances parameter efficiency. We then analyze how model size and sparsity ratios interact to identify optimal combinations.

**Experimental Setup.** Introducing network sparsity when scaling up model size requires careful control of multiple variables in our comparative experiments, including the width and depth of both actor and critic networks, as well as their respective sparsity levels. Since our primary goal

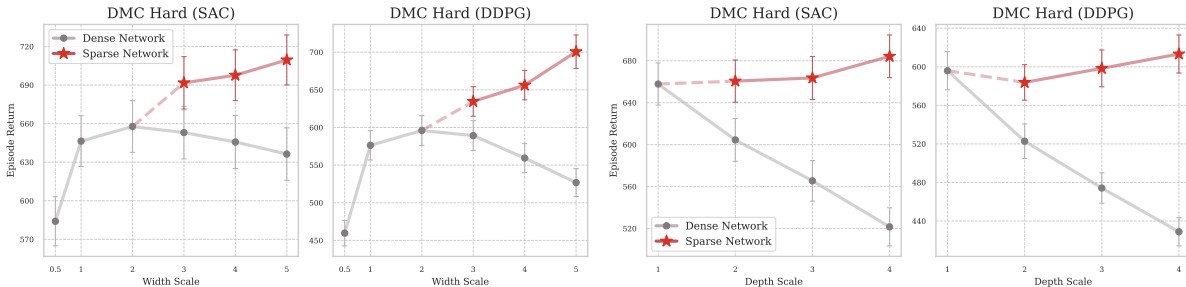

*Figure 2.* Network scaling experiments comparing dense and sparse SimBa architectures trained with SAC and DDPG on DMC Hard tasks. Results demonstrate that appropriate sparsity enables effective model scaling while preserving parameter efficiency.

is to investigate whether sparsity can extend DRL scaling boundaries beyond current dense model limitations, we establish the following experimental protocol to ensure fair and systematic evaluation: • Following SimBa's validated configuration, we maintain its default actor-critic size ratio where the critic network is four times wider and twice deeper than the actor network. • Taking the default SimBa model size as our baseline (actor hidden dimension of 128, critic hidden dimension of 512, one and two SimBa residual blocks for actor and critic respectively), we scale both networks by integer multiples in width or depth. For instance, a model with Width Scale = 2 and Depth Scale = 4 means the actor network uses two blocks with hidden dimension 256, while the critic network uses four blocks with hidden dimension 1024. • We specify only the overall sparsity level for both networks, with specific layer sparsity determined by the ER initialization. Complete experimental details are provided in Appendix B.1.

### 3.1. Appropriate Network Sparsity Both Enables Model Size Scaling and Enhances Parameter Efficiency

Although SimBa has integrated various recently proven interventions for mitigating DRL network pathologies and significantly improved its scalability, excessive model scaling still leads to performance collapse. As illustrated in Figure 2, when scaling the network along a single dimension beyond critical thresholds (exceeding 2× width or 1× depth of the baseline model), larger models yield worse performance. This degradation trend suggests that larger dense networks suffer from severely reduced parameter efficiency, preventing DRL agents from exploiting the theoretically increased representational capacity.

These findings motivate our first investigation: whether introducing appropriate sparsity during model scaling can break through current scaling limitations. Using the SimBa network with Width Scale = 2 and Depth Scale = 1 as the anchor point, we maintain the same learnable parameters count as the optimal dense model while increasing the total model size, exploring if such sparse scaling can more effectively leverage the increased network capacity.

The scaling trends for width and depth are presented in Figure 2, while Figure 1 additionally illustrates the combined

scaling of both dimensions. Detailed results for individual tasks can be found in Appendix B.2. First, when maintaining the same parameter count, larger sparse networks achieve superior performance compared to their smaller dense counterparts, indicating higher parameter efficiency. Second, at equal model sizes, sparse networks outperform their dense counterparts with fewer parameters, demonstrating more effective utilization of network capacity. Furthermore, in Section 4, we will demonstrate that these benefits fundamentally arise from appropriate network sparsity preventing DRL networks from falling into more severe optimization pathologies during scaling.

---

**Takeaway:** Weight-level sparsity, a simple architectural feature, can further unlock the scaling potential of DRL networks beyond the improved scalability achieved by advanced architectures, enabling networks to better harness both parameter efficiency and model capacity.

---

### 3.2. The Interplay between Model Size and Sparsity

Having established that appropriate network sparsity can enable better scaling, we next examine the practical implications of incorporating sparsity into DRL networks to derive effective implementation guidelines. Specifically, we treat model size and sparsity ratios as independent configuration parameters and analyze their performance impacts systematically. Several critical questions merit investigation, including: How does varying network sparsity influence the scaling potential of larger models? What role does sparsity play in enhancing default-sized model performance? Furthermore, what are the optimal combinations of model size and sparsity that yield the best DRL performance?

As shown in Figure 3, increasing network sparsity exhibits substantially different effects across model scales. For large networks, higher sparsity ratios consistently lead to improved performance across all tasks. This is particularly evident in challenging scenarios like *Humanoid Walk* and *Dog Run*, where large sparse networks achieve up to 40% higher returns compared to their dense counterparts. Such results strongly indicate that sparsity serves as an effective mechanism for unlocking the scaling potential of larger models. In contrast, default-sized networks show modest

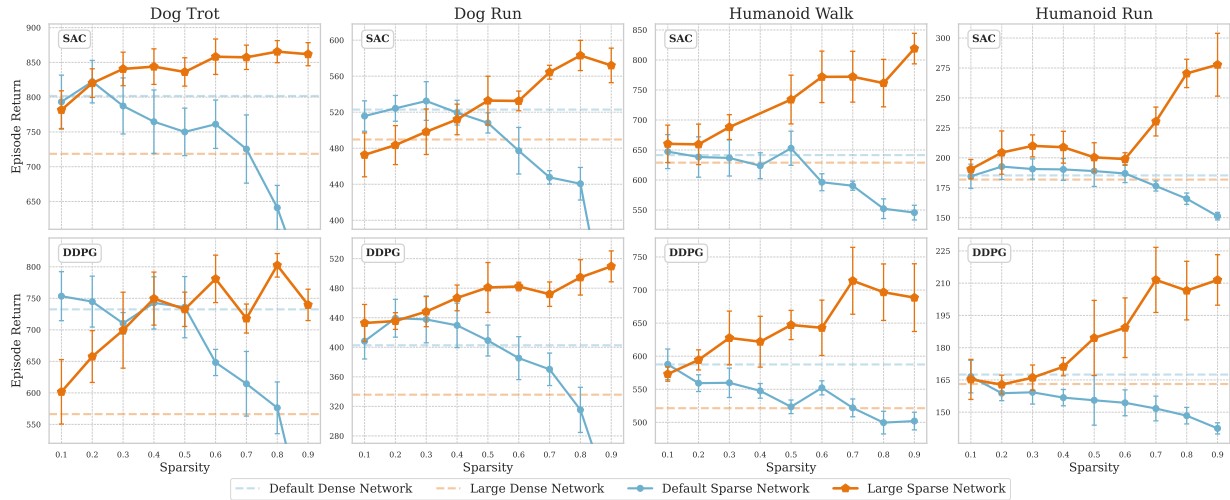

*Figure 3.* Scaling via network sparsity on four hardest DMC tasks using SAC and DDPG. For both default SimBa networks ($\sim 4.5$M parameters, blue lines) and large networks ($\sim 109$M parameters, orange lines), we systematically explore sparsity ratios from $0.1$ to $0.9$, with steps of $0.1$. Results demonstrate that while default networks suffer from high sparsity, large networks consistently benefit from increased sparsity ratios, highlighting the crucial role of sparsity in enabling effective model scaling.

performance gains with low sparsity ratios, validating the general benefits of network sparsity. However, their performance deteriorates significantly with higher sparsity, indicating that sufficient learnable parameters are essential for maintaining network expressivity.

> **Takeaway:** The best practice for scaling DRL networks is to increase model size while maintaining high static sparsity, achieving both efficient parameter utilization and superior representational expressivity.

## 4. Understanding the Barrier of Scaling and the Benefits of Network Sparsity

As demonstrated in the previous analysis, network sparsity empirically enhances DRL model scaling capabilities. This distinct scaling behavior raises the question: *how can sparse networks with fewer learnable parameters achieve superior performance compared to their dense counterparts?* To gain deeper insights into this phenomenon, we explore there critical factors: representation capacity, plasticity, regularization, and gradient interference. Our analysis reveals that while naively increasing DRL model size worsens optimization pathologies, introducing appropriate sparsity effectively mitigates these issues through multiple mechanisms.

### 4.1. Representational Capacity

The primary motivation for scaling up neural networks is to gain enhanced expressivity, allowing them to capture more complex relationships and learn more effective representations than their smaller counterparts. Therefore, we first investigate how sparsity impacts network capacity.

**Srank.** We characterize the representational capacity using the Stable-rank (Srank) ([Kumar et al., 2021](#)) of the critic

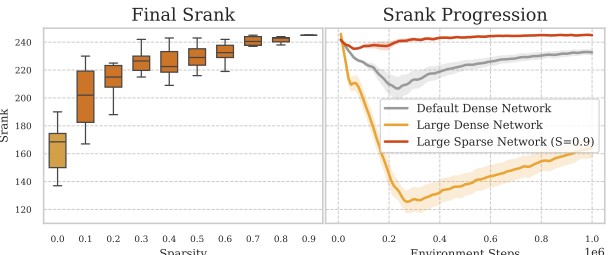

*Figure 4.* Analysis of network representation capacity via Srank metric on *Humanoid Run* using SAC. Network configurations match Figure 3. (*Left*) Final Srank after 1M steps across sparsity ratios. (*Right*) Srank progression for three network variants.

network. Srank measures the effective rank of learned representations, indicating the diversity and richness of the representations learned by the network. This is computed by performing eigenvalue decomposition of the feature matrix covariance and summing the indicators of singular values above a threshold $\tau$: $\mathrm{Srank} = \sum_{j=1}^{m} \mathbb{I}(\sigma_j > \tau)$, where $F \in \mathbb{R}^{d \times m}$ is the feature matrix containing $d$ samples of $m$-dimensional features, $\sigma_j$ denotes the singular values, and $\mathbb{I}(\cdot)$ is the indicator function.

As shown in Figure 4, our analysis reveals two key findings. First, increasing network sparsity leads to a consistent improvement in the critic's Srank, gradually approaching the theoretical upper bound of 256. More surprisingly, scaling up the network from $4.5$M to $109$M parameters leads to an unexpected degradation in representational capacity, reflected by a marked decline in Srank. This capacity collapse potentially explains the scaling barrier in DRL.

> **Takeaway:** Unlocking greater expressivity through scaling in DRL requires appropriate sparsity, as larger dense networks tend to suffer from severe capacity collapse.

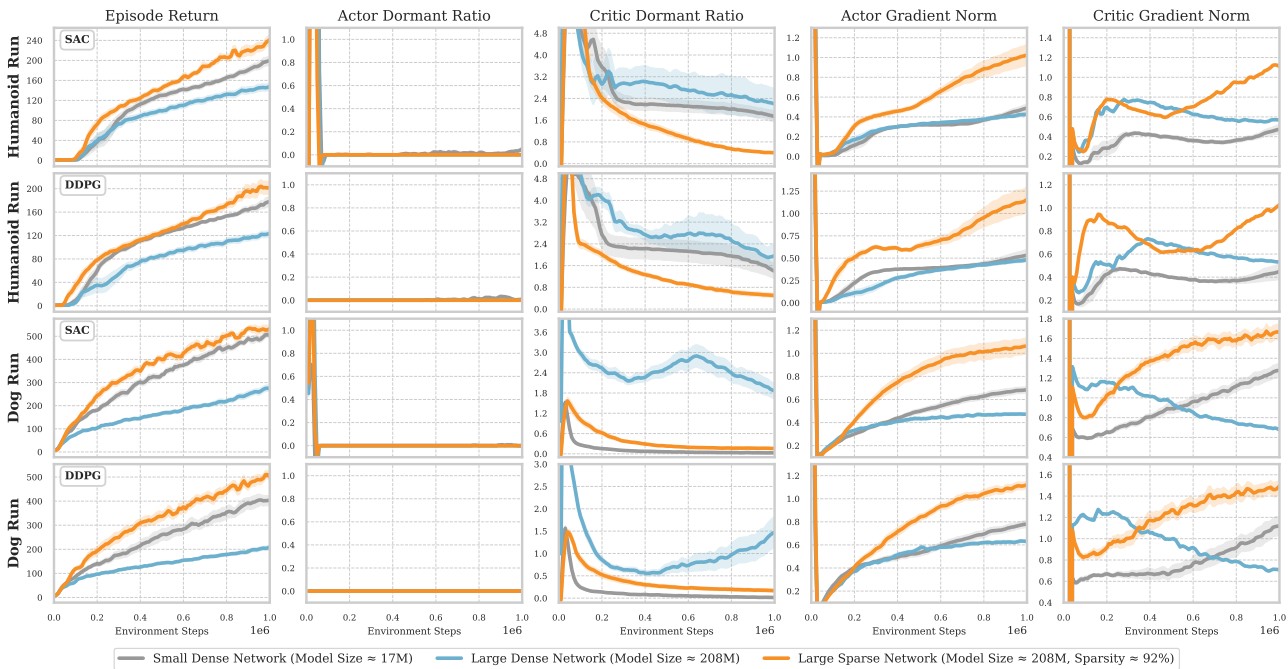

*Figure 5.* Plasticity measurements of three representative network configurations in the two most challenging tasks with SAC and DDPG. Despite employing the advanced SimBa architecture, large dense critic networks still suffer from rising neuron dormancy and gradient collapse as training progresses. Introducing sparsity, however, proves to be an effective solution, preventing such pathological trends.

## 4.2. Plasticity

Recent studies have identified plasticity loss as a key pathological symptom in DRL networks, where models progressively lose their ability to adapt to new experiences during training, eventually reaching learning stagnation (Nikishin et al., 2022; Lyle et al., 2023). While directly measuring plasticity remains challenging, several indicators have been found to strongly correlate with its deterioration, particularly the emergence of dormant neurons and the decay of gradient signals (Klein et al., 2024; Lewandowski et al., 2024). Hence, we characterize network plasticity through two key metrics: the dormant ratio (Sokar et al., 2023), which measures the proportion of inactive neurons, and gradient norm dynamics, which monitors the preservation of learning capability throughout training (Abbas et al., 2023).

**Dormant Ratio.** The dormant ratio is the proportion of dormant neurons within the entire network. Given an input distribution $\mathcal{D}$, A neuron $i$ in layer $\ell$ is considered dormant if its dormant score $\rho_i^\ell$ across input data $x \sim P(\cdot; \mathcal{D})$ falls below threshold $\tau$. The dormant score $\rho_i^\ell$ of an individual neuron can be defined as:

$$\rho_i^\ell = \frac{\mathbb{E}_{x \sim P(\cdot;\mathcal{D})}|h_i^\ell(x)|}{\frac{1}{H^\ell}\sum_{k \in h}\mathbb{E}_{x \sim P(\cdot;\mathcal{D})}|h_k^\ell(x)|} \quad (1)$$

where $h(x)$ denotes neuron activation and $H^\ell$ is the layer $\ell$ neuron count. A neuron $i$ in layer $\ell$ is $\tau$-dormant if $\rho_i^\ell \leq \tau$.

**Gradient Norm.** We monitor the L2 norm of network gradients over active (non-pruned) parameters to quantify

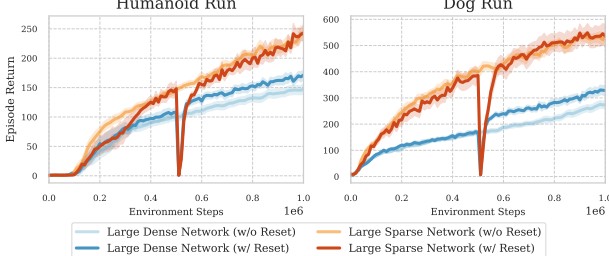

*Figure 6.* Reset diagnostic comparison for large dense networks and large sparse networks. Despite dense networks relying on Reset operations to recover plasticity, sparse networks maintain learning capability naturally without such remedial interventions.

the strength of learning signals during training, where diminishing gradient norms potentially signal a loss of plasticity.

The distinct plasticity dynamics between sparse and dense networks are illustrated in Figure 5. Benefiting from the advanced SimBa architecture, small dense networks maintain healthy plasticity throughout training without significant deterioration. However, as model size increases, large dense networks exhibit clear signs of plasticity loss, manifested through rising critic dormant ratios and collapsing gradient norms in later training stages. Notably, actor networks show minimal plasticity deterioration, aligning with the findings in Ma et al. (2024). In contrast, large sparse networks effectively mitigate the severe plasticity loss commonly observed in larger models. Moreover, using equivalent parameter counts, large sparse networks achieve comparable or lower dormant ratios than small dense networks while sustaining stronger gradient signals throughout training.

**Reset as a Diagnostic Tool.** Given Reset serves as a direct approach to restore plasticity in DRL networks (Nikishin et al., 2022), we employ this operation to assess whether plasticity loss remains a critical issue in the large sparse SimBa network. As shown in Figure 6, Reset operations significantly boost the performance of large dense networks by breaking their learning stagnation yet provide no benefits to large sparse networks, and may even slightly harm performance by disrupting established training dynamics.

In past practices, scaling up vanilla MLP networks often required Reset operations. With the progressive introduction of architectural advances, this dependence on Reset gradually diminished (Lee et al., 2024). Now, by incorporating sparsity into SimBa and simultaneously increasing both model size and sparsity ratio, we have completely eliminated the need for Reset while achieving superior scalability.

> **Takeaway:** Weight-level sparsity effectively preserves plasticity in large-scale networks, eliminating plasticity loss as a bottleneck for scaling up SimBa architectures.

### 4.3. Regularization

We then examine how network sparsity serves as an implicit regularization mechanism that simultaneously constrains weight magnitudes and induces beneficial inductive biases towards simpler solutions.

**Parameter Norm.** Unbounded parameter growth has been identified as a critical pathological behavior in deep RL networks, leading to training instability and severely hindering the network's ability to effectively learn value functions and policies (Lyle et al., 2024b).

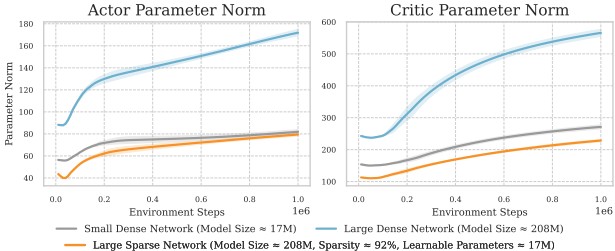

*Figure 7.* Parameter norm evolution for actor and critic networks, corresponding to the *Humanoid Run* (SAC) scenario in Figure 5.

While network sparsification inherently reduces the total number of parameters compared to dense architectures, a remarkable finding emerges in Figure 7: large sparse networks exhibit similar or even lower parameter norms compared to small dense networks with equivalent learnable parameters. This suggests that sparsity serves as an effective implicit regularizer beyond mere parameter reduction.

**Simplicity Bias.** Neural networks inherently favor learning simpler patterns over complex ones, a phenomenon known as simplicity bias (Shah et al., 2020; Berchenko, 2024). To quantify this bias, we utilize the simplicity bias score

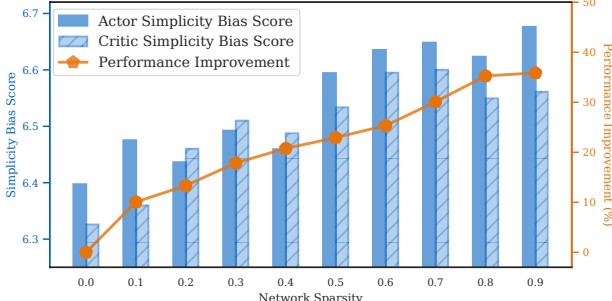

*Figure 8.* Simplicity bias scores and performance improvements across different sparsity ratios, where performance gains are averaged over large networks in eight scenarios from Figure 5.

from Lee et al. (2024) that evaluates network complexity at initialization to avoid confounding factors from the non-stationary RL training dynamics.

Figure 8 demonstrates that network sparsity consistently promotes higher simplicity bias scores, correlating with improved performance in scaled-up architectures. Remarkably, SimBa's comprehensive architectural improvements yielded only a 0.5 increase in simplicity bias scores (from 5.8 to 6.3) (Lee et al., 2024), while our simple one-shot pruning approach further raises these scores by 0.3-0.4, highlighting the effectiveness of sparsity as a key network property.

> **Takeaway:** Sparsity is an effective regularizer that can control parameter growth and promote simpler solutions.

### 4.4. Gradient Interference

Finally, we investigate whether network sparsity affects the interactions between gradients from different data points - a phenomenon called gradient interference that impacts learning dynamics (Bengio et al., 2020; Lyle et al., 2022b). Following the analytical approach in Lyle et al. (2023), we will estimate the interference level by gradient covariance matrices, which are computed by sampling $k$ training points $\mathbf{x}_1, \dots, \mathbf{x}_k$, and constructing $C_k \in \mathbb{R}^{k \times k}$ with entries:

$$C_k[i, j] = \frac{\langle \nabla_\theta \ell(\theta, \mathbf{x}_i), \nabla_\theta \ell(\theta, \mathbf{x}_j) \rangle}{\|\nabla_\theta \ell(\theta, \mathbf{x}_i)\| \|\nabla_\theta \ell(\theta, \mathbf{x}_j)\|} \quad (2)$$

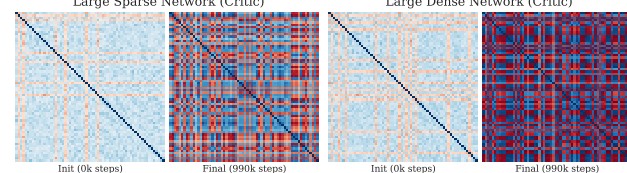

*Figure 9.* Gradient covariance matrices of large sparse and dense networks before and after training. Darker blue indicates strongly aligned gradients, darker red indicates strongly conflicting gradients, and lighter colors indicate more independent gradients. Sparse networks maintain more independent (less interfering) gradients throughout training compared to dense networks.

Figure 9 shows that while sparse and dense networks start with similar gradient correlations, sparse networks maintain

significantly weaker correlations throughout training.

> **Takeaway:** Network sparsity naturally promotes gradient orthogonality, mitigating gradient interference.

## 5. Sparsity Boosts Scaling in Broader Setups

To examine the broader applicability of our findings, we extend our experiments to visual RL and streaming RL scenarios. Here we report core results, with complete experimental details provided in Appendix C.

**Visual RL.** In our visual RL experiments, we evaluate on image-based DMC with DrQ-v2 (Yarats et al., 2022) as our baseline. Building on the insights of Ma et al. (2024), we only scale the critic network while keeping the actor fixed.

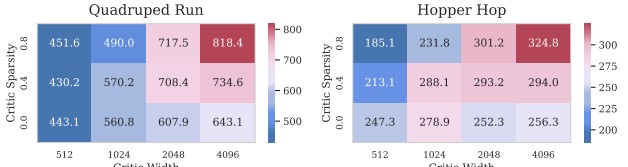

*Figure 10.* Scaling via network sparsity and critic width on two representative visual RL tasks, reporting mean episode returns averaged over 5 random seeds after 2M environment steps.

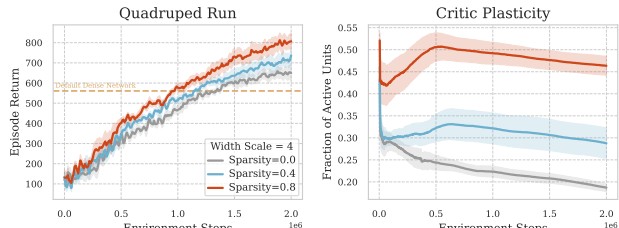

*Figure 11.* Learning curves and critic plasticity for Quadruped Run under varying sparsity levels with a 4x wider critic network.

Results in Figure 10 show that while increasing critic width leads to gradual improvements with dense networks, combining network sparsity (0.8) with larger widths can further boost performance significantly, highlighting the effectiveness of sparsity in visual RL scaling. We dive into the plasticity dynamics to understand these performance differences. As shown in Figure 11, scaling up the critic width by 4x leads to severe plasticity deterioration in dense networks, where a large fraction of neurons become inactive. However, introducing sparsity effectively mitigates this issue, aligning with our findings in Section 4.2 that weight-level sparsity helps preserve neuron-level plasticity throughout training.

**Streaming RL.** Streaming RL agents process each sample immediately upon arrival without storing past experiences, exacerbating the non-stationarity of the learning process and resulting in significant sample inefficiency (Elsayed et al., 2024; Vasan et al., 2024). To overcome this stream barrier, Elsayed et al. (2024) propose SparseInit by randomly initializing most weights to zeros. Unlike our one-shot pruning at initialization which maintains fixed sparsity throughout

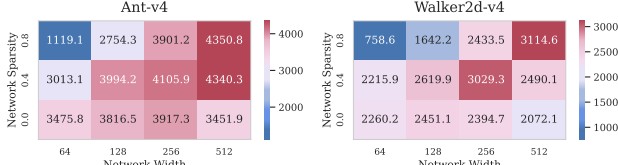

*Figure 12.* Streaming RL network scaling performance across sparsity ratios and widths, averaged over 5 seeds after 2M steps.

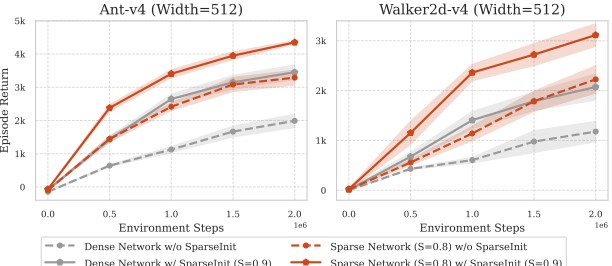

*Figure 13.* Comparing static sparsity (fixed pruned weights) and SparseInit (trainable zero-initialized weights) in streaming RL.

training, SparseInit only zeroes weights at initialization, allowing gradient updates to gradually reduce sparsity during training. Beyond showing that static pruning enables effective network scaling in streaming RL (Figure 12), we demonstrate that both one-shot pruning and SparseInit significantly improve sample efficiency (Figure 13), underlining the broad benefits of network sparsity in RL training.

## 6. Conclusion

This work demonstrates that ***static network sparsity***, achieved through one-shot random pruning, is a simple yet powerful tool for unlocking the scaling potential of DRL. By carefully studying its effects on representational capacity (via Srank), simplicity bias, and gradient interference, we show that sparsity not only mitigates optimization pathologies but also enables larger models to achieve superior performance across diverse RL settings, including challenging streaming RL scenarios. These findings are particularly significant because they address a long-standing challenge in RL: scaling models effectively, an area where RL has historically struggled compared to supervised learning.

Our approach is easy to implement and readily compatible with any RL algorithm, requiring only a one-time preprocessing step. Unlike dynamic methods which update sparsity patterns during training, static sparsity maintains a fixed sparse structure throughout training, avoiding both training instability and additional computational overhead.

Our results highlight the importance of architectural choices in DRL and suggest that network architecture and RL algorithms should not be studied in isolation. By establishing sparsity as a key enabler of scalability, this work opens new avenues for research into specialized sparsity structures, dynamic sparsity methods, and theoretical frameworks to make RL more practical and deployable in real-world settings.

## Acknowledgment

This project is supported by the National Research Foundation, Singapore, under its NRF Professorship Award No. NRF-P2024-001. Lu Li and Pierre-Luc Bacon are supported by CIFAR. This research is enabled in part by compute resources, software and technical help provided by Mila (mila.quebec).

## Impact Statement

This paper presents work whose goal is to advance the field of Machine Learning. There are many potential societal consequences of our work, none which we feel must be specifically highlighted here.

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

The appendix is divided into several sections, each giving extra information and details.

## A. Related Work

In this section, we review several topics closely related to our work. We begin by discussing the optimization pathologies unique to deep reinforcement learning (DRL). We then examine the scaling barriers in DRL models and various attempts to enhance their scalability. Additionally, we review existing applications of sparse networks in DRL, whether aimed at model compression, training acceleration, or performance enhancement.

### A.1. Optimization Pathologies in DRL

Although deep neural networks have driven remarkable advances in current deep reinforcement learning (DRL) applications, growing evidence indicates that DRL networks are prone to severe optimization pathologies during training (Nikishin, 2024; Nauman et al., 2024a; Goldie et al., 2024). These pathologies emerge from the unique challenges of integrating DL mechanisms with the RL paradigm, presenting distinctive issues not encountered in traditional RL settings. Such difficulties stem from fundamental characteristics of reinforcement learning that distinguish it from supervised learning: non-stationary data distributions and optimization objectives, as well as the inherent nature of learning through online interactions.

These DRL-specific pathologies have recently been identified and characterized through various phenomena: primacy bias (Nikishin et al., 2022), the dormant neuron phenomenon (Sokar et al., 2023), implicit under-parameterization (Kumar et al., 2021), capacity loss (Lyle et al., 2022a), and the broader issue of plasticity loss (Klein et al., 2024; Abbas et al., 2023; Juliani & Ash, 2024). Although these studies approach the problem from different angles, they converge on a common finding: DRL networks routinely develop severe optimization pathologies during training that fundamentally impair their ability to learn from new experiences. The consequences manifest either as severe sample inefficiency or, in the worst cases, complete learning stagnation (Ma et al., 2024). These pathologies manifest through several observable symptoms: a high proportion of inactive neurons (Sokar et al., 2023), reduced effective rank of representational features (Kumar et al., 2021; Lyle et al., 2022a), unbounded growth in parameter norms (Lyle et al., 2023), and increased gradient interference across training samples (Lyle et al., 2024b). Each of these symptoms contributes to the network's diminishing ability to effectively learn and optimize both the policy and value functions.

Moreover, such pathological behaviors intensify with increasing model size, fundamentally limiting the scaling capabilities of DRL models (Ceron et al., 2024a;b; Bjorck et al., 2021). Consequently, a critical bottleneck unique to DRL has emerged: *how to effectively scale up neural networks for better representation capacity while avoiding falling into severe optimization pathologies.* In this work, we demonstrate a surprisingly simple yet powerful alternative: static network pruning prior to training. Through extensive experimental comparisons and empirical analysis, we show that this approach to network

sparsity effectively unlocks the scaling potential of DRL networks while significantly mitigating optimization pathologies.

## A.2. Network Scaling in DRL

Using deep neural networks is a key factor in successfully applying reinforcement learning to complex tasks. However, while some recent advances in supervised learning have been driven by scaling up the number of network parameters, a phenomenon commonly referred to as *scaling laws*, it remains challenging to increase the number of parameters in deep reinforcement learning without experiencing performance degradation. Several recent works in DRL have addressed this by scaling up network sizes through various strategies. Schwarzer et al. (2023) transitioned from the original CNN architecture to the ResNet-based Impala-CNN architecture (Espeholt et al., 2018) and scaled the network width by a factor of 4. Both BRO (Nauman et al., 2024b) and SimBa (Lee et al., 2024) employed deeper networks that incorporate layer normalization (Lei Ba et al., 2016) and residual connections. Ceron et al. (2024b) incorporated a soft Mixture-of-Experts module (Puigcerver et al., 2020) into value-based networks, resulting in more parameter-scalable models and improved performance. Farebrother et al. (2024) show that value functions trained using categorical cross-entropy substantially enhance performance and scalability in multiple domains. Ceron et al. (2024a) utilized magnitude pruning on value-based networks, progressively decreasing the number of parameters in dense architectures during training to achieve highly sparse models, leading to improved performance when scaling network width. Despite these advances, scaling up network sizes in DRL using random static sparsity remains underexplored.

## A.3. Sparse Networks in DRL

Initial explorations of sparse networks in DRL were primarily motivated by the potential for model compression, aiming to accelerate training and facilitate efficient model deployment (Tan et al., 2023). Early explorations of network sparsification in DRL primarily focused on behavior cloning and offline RL settings (Arnob et al., 2021; Vischer et al., 2021). In the more challenging context of online RL, Sokar et al. (2021) explored the application of Sparse Evolutionary Training (SET) (Mocanu et al., 2018) and successfully achieved 50% sparsity. However, attempts to increase sparsity beyond this level resulted in significant training instability. Subsequently, Tan et al. (2023) enhanced the efficacy of dynamic sparse training through a novel delayed multi-step temporal difference target mechanism and a dynamic-capacity replay buffer, ultimately achieving sparsity levels of up to 95%. Graesser et al. (2022) conducted a comprehensive investigation and demonstrated that pruning consistently outperforms standard dynamic sparse training methods, such as SET (Mocanu et al., 2018) and RigL (Evci et al., 2020). Data Adaptive Pathway Discovery (DAPD) (Arnob et al., 2024) dynamically adjusts network pathways in response to online RL distribution shifts, maintaining effectiveness at high sparsity levels.

Beyond the initial goal of achieving parameter-efficient architectures through sparsity, recent studies have recognized that sparse and adaptive networks can enhance DRL model scalability while mitigating training pathologies such as plasticity loss. For instance, Ceron et al. (2024a) shows that applying gradual magnitude pruning to large models significantly enhances the performance of value-based agents. Similarly, Ceron et al. (2024b) demonstrates that incorporating Soft MoEs into value-based RL networks enables better parameter scaling. Furthermore, Neuroplastic Expansion (Liu et al., 2024) addresses plasticity challenges by progressively evolving networks from sparse to dense architectures, effectively leveraging increased model capacity. Although these approaches differ in implementation, they all fall under the broader category of dynamic sparse training, where network topology evolves during training. In contrast, this work isolates sparsity as a standalone feature, revealing that static sparse training through random pruning at initialization alone can substantially enhance DRL network scalability, addressing a significant gap in current research.

# B. Detailed Experimental Setup and Results of Main Experiments

This section details the experimental setup and the results of our evaluation in Section 3 and Section 4. **The code is available in the supplementary materials**.

## B.1. Detailed Experimental Setup

We evaluate SAC and DDPG with SimBa network with varying sizes and sparsity levels on 6 hard tasks of DeepMind Control Suites (Tassa et al., 2018), also known as **DMC Hard**. The complete list for **DMC Hard** is provided in Table 1. Note that we omit Dog Stand from this set since the default SimBa architecture already demonstrates strong performance on this task, consistently achieving scores above 900 on the normalized 1000-point scale.

*Table 1.* **DMC Hard** consists of 6 continuous control tasks.

| Task | Observation dim | Action dim |
|------|-----------------|------------|
| Dog Run | 223 | 38 |
| Dog Trot | 223 | 38 |
| Dog Walk | 223 | 38 |
| Humanoid Run | 67 | 24 |
| Humanoid Stand | 67 | 24 |
| Humanoid Walk | 67 | 24 |

The experimental settings for DMC in Section 3 and Section 4 are primarily adapted from those employed in SimBa. Most of the hyperparameters in our experiments are identical to those used in Lee et al. (2024), except for the network width (hidden dimension) and depth (number of blocks), as detailed in Table 2 and Table 3. Unless otherwise specified, all experiments are conducted using 8 random seeds.

*Table 2.* **SAC hyperparameters.** The hyperparameters listed below are used consistently across all experiments in Section 3 and Section 4. For the discount factor, we follow Lee et al. (2024) using heuristics used by TD-MPC2 (Hansen et al., 2023).

| Hyperparameter | Value |
|----------------|-------|
| Critic block type | SimBa |
| Critic num blocks | $\{2,4,6,8\}$ |
| Critic hidden dim | $\{256,512,1024,1536,2048,2560\}$ |
| Critic learning rate | 1e-4 |
| Target critic momentum ($\tau$) | 5e-3 |
| Actor block type | SimBa |
| Actor num blocks | $\{1,2,3,4\}$ |
| Actor hidden dim | $\{64,128,256,384,512,640\}$ |
| Actor learning rate | 1e-4 |
| Initial temperature ($\alpha_0$) | 1e-2 |
| Temperature learning rate | 1e-4 |
| Target entropy ($\mathcal{H}^*$) | $|\mathcal{A}|/2$ |
| Batch size | 256 |
| Optimizer | AdamW |
| Optimizer momentum ($\beta_1, \beta_2$) | (0.9, 0.999) |
| Weight decay ($\lambda$) | 1e-2 |
| Discount ($\gamma$) | Heuristic |
| Replay ratio | 2 |
| Clipped Double Q | False |

*Table 3.* **DDPG hyperparameters.** The hyperparameters listed below are used consistently across all experiments in Section 3 and Section 4. For the discount factor, we follow Lee et al. (2024) using heuristics used by TD-MPC2 (Hansen et al., 2023).

| Hyperparameter | Value |
|---|---|
| Critic block type | SimBa |
| Critic num blocks | {2,4,6,8} |
| Critic hidden dim | {256,512,1024,1536,2048,2560} |
| Critic learning rate | 1e-4 |
| Target critic momentum ($\tau$) | 5e-3 |
| Actor block type | SimBa |
| Actor num blocks | {1,2,3,4} |
| Actor hidden dim | {64,128,256,384,512,640} |
| Actor learning rate | 1e-4 |
| Exploration noise | $\mathcal{N}(0, 0.1^2)$ |
| Batch size | 256 |
| Optimizer | AdamW |
| Optimizer momentum ($\beta_1, \beta_2$) | (0.9, 0.999) |
| Weight decay ($\lambda$) | 1e-2 |
| Discount ($\gamma$) | Heuristic |
| Replay ratio | 2 |
| Clipped Double Q | False |

## B.2. Detailed DMC Results

**Scaling Trends Visualization.** Figure 14 presents an alternative visualization of the model scaling results shown in Figure 2, using a linear scale for model size instead of the logarithmic scale used in the main text. This alternative visualization emphasizes the widening performance gap between sparse and dense networks, which becomes particularly pronounced at larger model scales (>100M parameters).

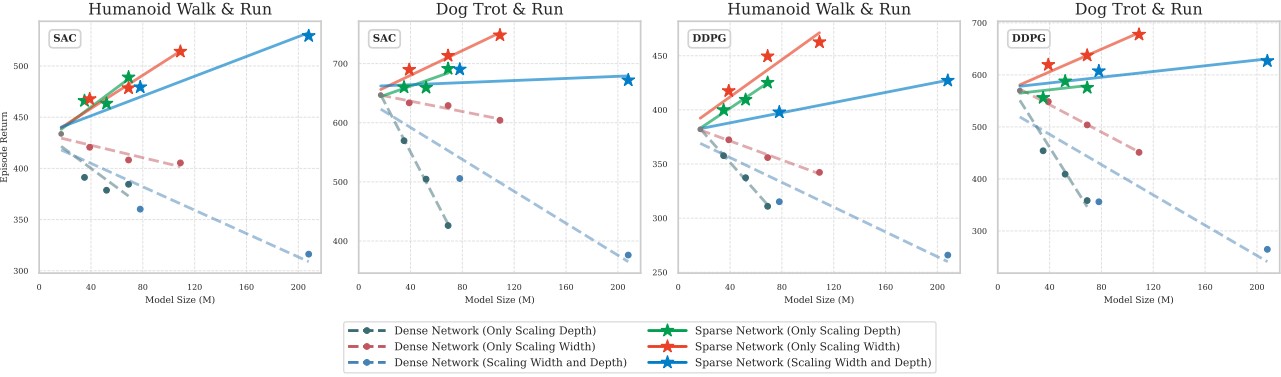

*Figure 14.* Model scaling trends of ★★★**sparse** versus ●●●**dense** networks on four hardest DMC tasks using SimBa architecture with SAC and DDPG. The data points in this figure are identical to those shown in Figure 1; however, this figure employs a linear scale for model size, providing an alternative view of the scaling relationships.

**Single Task Results.** We provide a detailed breakdown of the scaling trends for individual tasks that were aggregated in Figure 2. The width scaling results are presented in Figure 15, while the depth scaling results are shown in Figure 16. For each task, we use the optimal dense network as a reference point and explore larger model sizes with appropriate sparsity levels to maintain constant parameter counts. The consistency between individual task trends (Figure 15 and Figure 16) and the aggregated results (Figure 2) reinforces the generality of our findings.

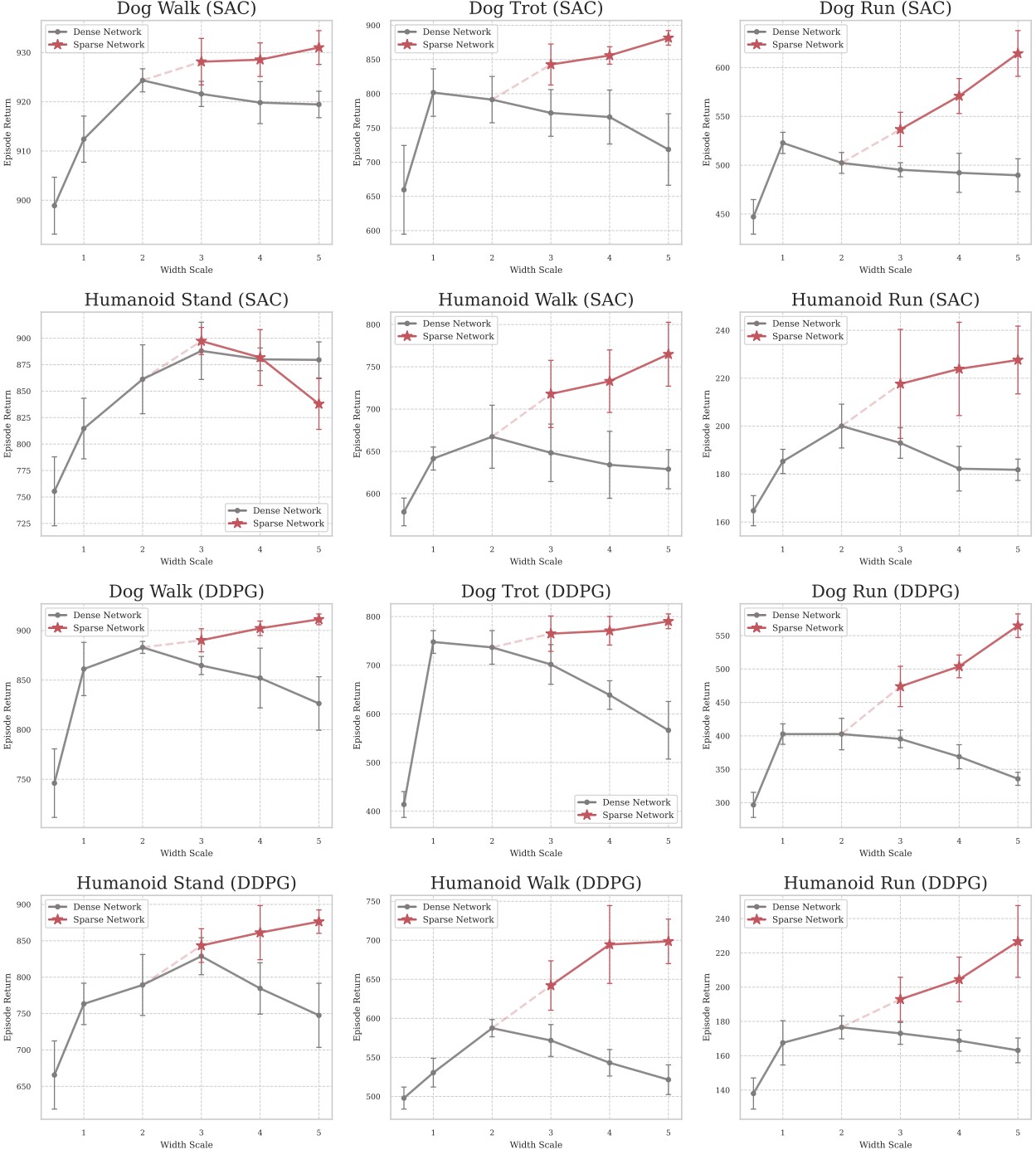

*Figure 15.* Width scaling experiments comparing dense and sparse networks across all DMC Hard tasks. Results show episode returns for both SAC (top two rows) and DDPG (bottom two rows) implementations on six challenging control tasks. Each data point represents the mean performance across 8 random seeds, with error bars indicating standard deviation.

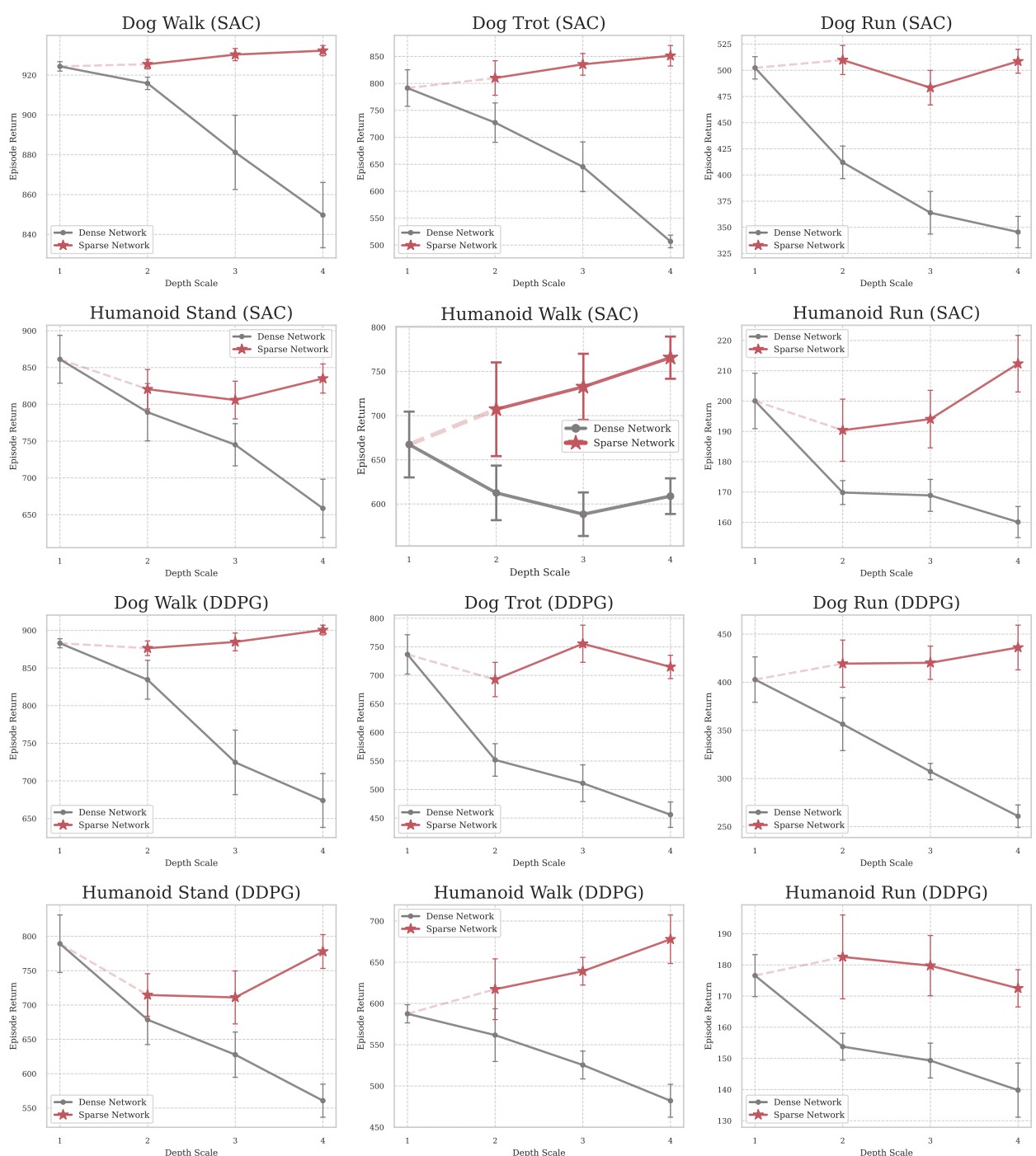

*Figure 16.* Depth scaling experiments comparing dense and sparse networks across all DMC Hard tasks. Results show episode returns for both SAC (top two rows) and DDPG (bottom two rows) implementations on six challenging control tasks. Each data point represents the mean performance across 8 random seeds, with error bars indicating standard deviation.

## C. Detailed Experimental Setup of broader setups

### C.1. Visual RL

We conducted visual RL experiments on DMC using image input as the observation. All experiments were based on the DrQ-v2 (Yarats et al., 2022), with all hyperparameters retained from the original DrQ-v2 implementation. The sole modification involved adjusting the width of the critic network to accommodate specific experimental settings. The hyperparameters are presented in Table 4.

In Figure 11, we use the Fraction of Active Units (FAU) as a metric for measuring plasticity. The FAU for neurons located in module $\mathcal{M}$, denoted as $\Phi_{\mathcal{M}}$, is formally defined as:

$$\Phi_{\mathcal{M}} = \frac{\sum_{n \in \mathcal{M}} \mathbf{1}(a_n(x) > 0)}{N}, \tag{3}$$

where $a_n(x)$ represent the activation of neuron $n$ given the input $x$, and $N$ is the total number of neurons within module $\mathcal{M}$.

*Table 4.* **DrQ-v2 hyperparameters.**

| Hyperparameter | Value |
|---|---:|
| Replay buffer capacity | $10^6$ |
| Action repeat | 2 |
| Seed frames | 4000 |
| Exploration steps | 2000 |
| $n$-step returns | 3 |
| Mini-batch size | 256 |
| Discount $\gamma$ | 0.99 |
| Optimizer | Adam |
| Learning rate | $10^{-4}$ |
| Critic Q-function soft-update rate $\tau$ | 0.01 |
| Features dim. | 50 |
| Repr. dim. | $32 \times 35 \times 35$ |
| Hidden dim. | 1024 |
| Exploration stddev. clip | 0.3 |
| Exploration stddev. schedule | $\mathrm{linear}(1.0, 0.1, 500000)$ |

## C.2. Streaming RL

We conducted streaming RL experiments on two MuJoCo robot locomotion tasks (Todorov et al., 2012), Ant-v4 and Walker2d-v4. All experiments were based on the Stream AC($\lambda$) algorithm (Elsayed et al., 2024), with all hyperparameters retained from the original Stream AC($\lambda$) implementation. The sole modification involved adjusting the width of the actor network and the critic network to accommodate specific experimental settings. The learning curves for agents with different network widths and sparsity levels are presented in Figure 17.

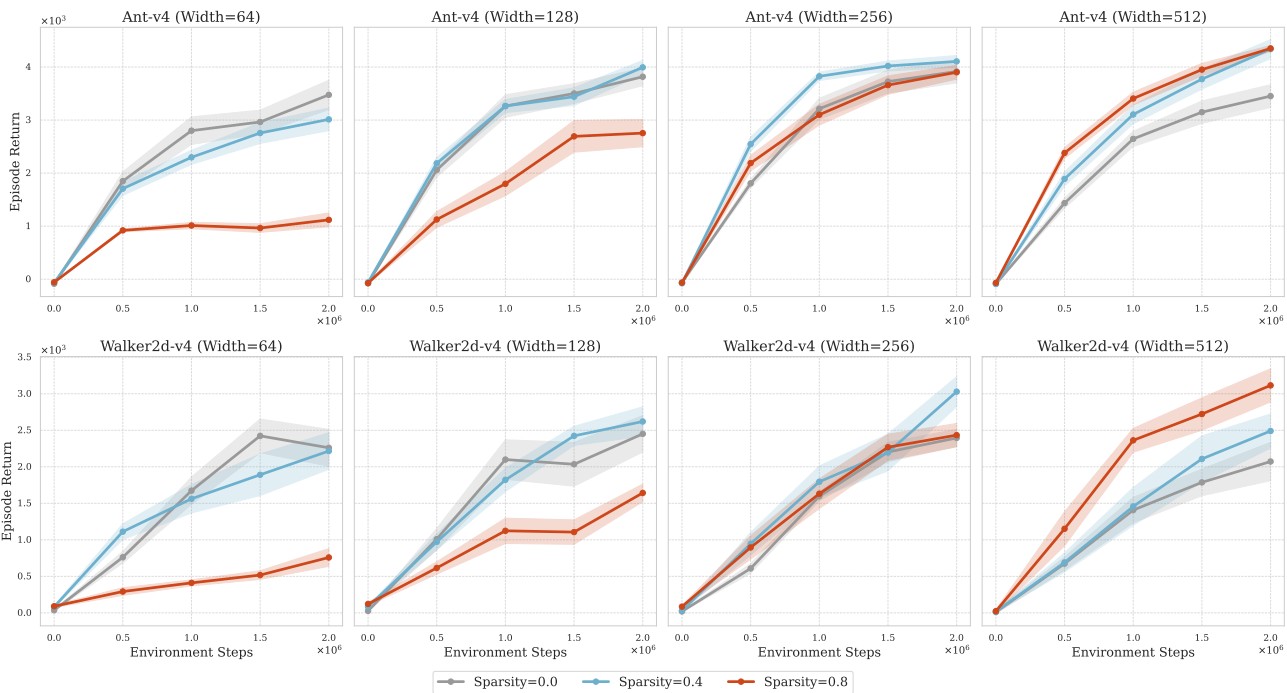

*Figure 17.* Learning curves of Stream AC($\lambda$) agent on Ant-v4 and Walker2d-v4, evaluated across varying sparsity levels and network widths for both the actor and critic networks.

## C.3. Atari-100k

We conducted Atari experiments on the Atari-100k benchmark (Kaiser et al., 2020), where the agent may perform only 100K environment steps, roughly equivalent to two hours of human gameplay. Our experiments were based on Data Efficient Rainbow (DER) (Van Hasselt et al., 2019), a variant of Rainbow (Hessel et al., 2018) tuned for sample efficiency. All experiments were based on Dopamine (Castro et al., 2018), except that we used an IMPALA CNN architecture (Espeholt et al., 2018) instead of NatureCNN. The hyperparameters are presented in Table 5.

We increase the width of the IMPALA CNN by a factor of three and evaluate three static sparsity configurations: dense (0.0), moderate sparsity (0.4), and high sparsity (0.8). Figure 18 shows the improvement relative to the default setting, which uses the default size and dense network. As in previous experiments, the results indicate that introducing static sparsity into DRL networks can unlock their scaling potential and yield performance improvements. We note that the Atari-100k low-data regime may not fully demonstrate the benefits of scaling, and more comprehensive studies with longer training would be valuable for future work.

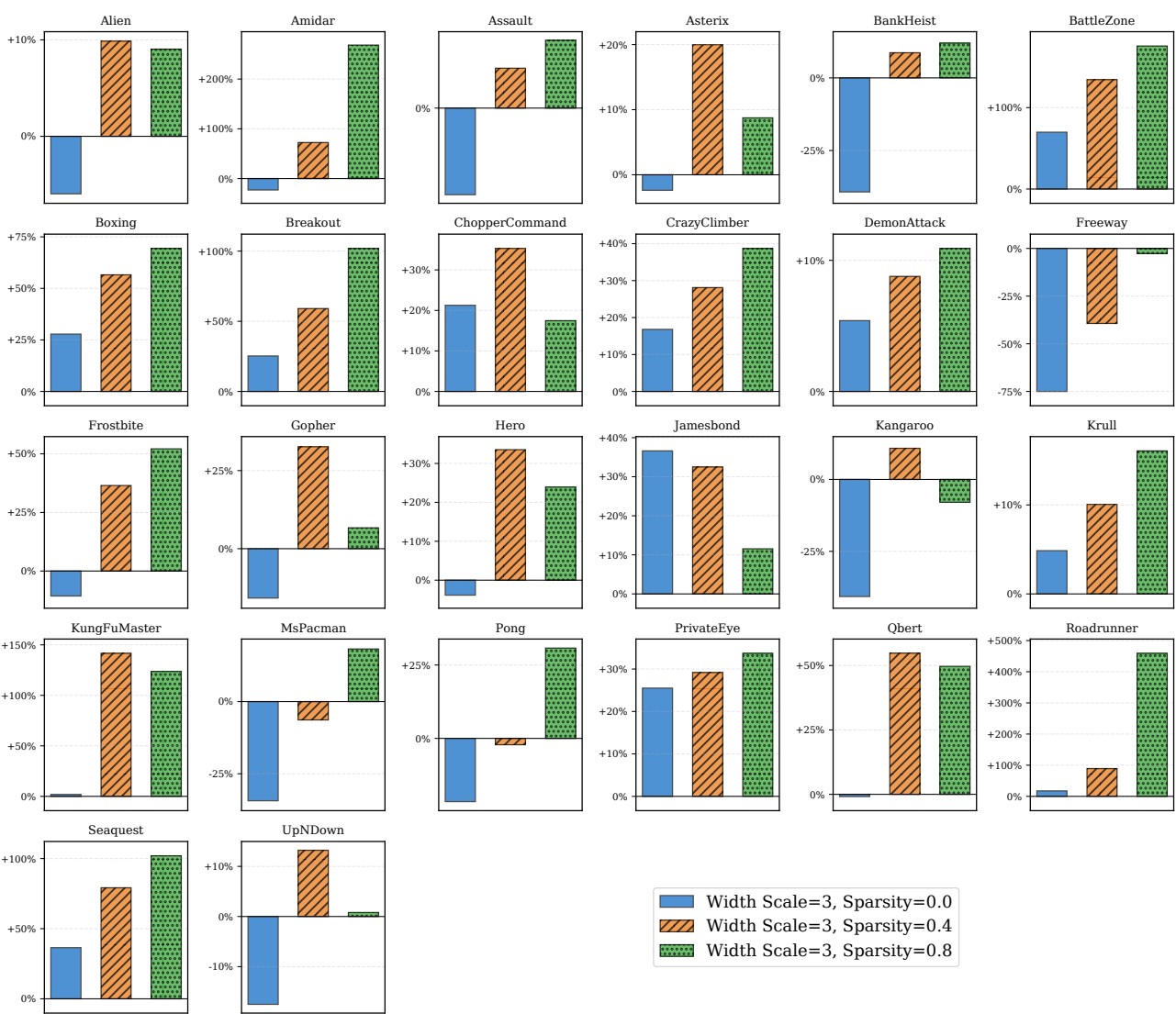

*Figure 18.* Performance improvements on Atari-100k benchmark when scaling network width (3x default size) with different sparsity levels using Data Efficient Rainbow (DER). Each bar represents the percentage improvement relative to the default dense network configuration. The results demonstrate that introducing sparsity (both at 0.4 and 0.8 levels) generally yields better performance than dense models (0.0, blue) when scaling model size, with optimal sparsity levels varying across different games. This extends our findings from continuous control domains to discrete action spaces, suggesting that the benefits of network sparsity are robust across different reinforcement learning environments.

Table 5. **DER hyperparameters.**

| Hyperparameter | Value |
|---|---:|
| Gray-scaling | True |
| Observation down-sampling | 84x84 |
| Frames stacked | 4 |
| Action repetitions | 4 |
| Reward clipping | [-1, 1] |
| Terminal on loss of life | True |
| Update | Distributional Q |
| Dueling | True |
| Support of Q-distribution | 51 |
| Discount factor | 0.99 |
| Minibatch size | 32 |
| Optimizer | Adam |
| Optimizer: learning rate | 0.0001 |
| Optimizer: $\epsilon$ | 0.00015 |
| Exploration | Noisy nets |
| Noisy nets parameter | 0.5 |
| Training steps | 100K |
| Evaluation trajectories | 100 |
| Min replay size for sampling | 1600 |
| Updates per step | 1 |
| Multi-step return length | 10 |
| CNN network | IMPALA CNN (Espeholt et al., 2018) |
| Target network update period | 2000 |

