# OpenReview forum: "Network Sparsity Unlocks the Scaling Potential of Deep Reinforcement Learning"
_ICML.cc/2025/Conference — ICML 2025 oral_

### Official Review · Reviewer_mWRx · 2025-03-12

**Overall Recommendation:** 4

**Summary:**

The paper builds upon previous work on scaling model size in RL [1], and extends its limit with simple layer-wise random pruning at initialization [2]. This is primarily verified in state-based RL (DMC), where a pruned large network greatly surpasses a dense network with the same number of trainable parameters. This is further extended to vision-based RL and streaming RL setting, which all have shown similar tendencies. Finally, they provide an extensive analysis on the benefits of one-shot pruning, specifically in the regime of representation capacity (s-rank), plasticity (dormant ratio, gradient norm, parameter norm), and gradient inference.

[1] SimBa: Simplicity Bias for Scaling Up Parameters in Deep Reinforcement Learning., ICLR'25.

[2] The Unreasonable Effectiveness of Random Pruning: Return of the Most Naive Baseline for Sparse Training., ICLR'22.

**Claims And Evidence:**

The effectiveness of one-shot pruning has been shown by prior work: The Unreasonable Effectiveness of Random Pruning: Return of the Most Naive Baseline for Sparse Training (ICLR'22).

**Essential References Not Discussed:**

I think most references in mind were present. However, the works on iterative pruning could be included:
1. In deep reinforcement learning, a pruned network is a good network., ICML’24.
2. Data-Efficient GAN Training Beyond (Just) Augmentations: A Lottery Ticket Perspective., NeurIPS’21.

**Experimental Designs Or Analyses:**

Authors clearly show the benefits of one-shot pruning with metrics commonly used in diverse fields:

1. Higher s-rank (larger representation capacity)
2. Lower dormant ratio (better plasticity preservation [1])
3. Larger gradient norm (better plasticity preservation)
4. Smaller parameter norm (better plasticity preservation [2])
5. Reset doesn’t help (proof of plasticity preservation).
6. Higher simplicity bias score [3]
7. Gradients closer to orthogonal (less gradient inference [4])

[1] The Dormant Neuron Phenomenon in Deep Reinforcement Learning., ICML'23.

[2] Normalization and effective learning rates in reinforcement learning., NeurIPS'24.

[3] SimBa: Simplicity Bias for Scaling Up Parameters in Deep Reinforcement Learning., ICLR'25.

[4] Understanding Plasticity in Neural Networks., ICML'23.

**Methods And Evaluation Criteria:**

The proposed method (one-shot pruning) is well verified to 'unlock parameter scaling in RL', as shown by the consistent improvement in challenging DMC Hard tasks, which prior work (Simba) fails to follow.

**Other Comments Or Suggestions:**

Overall, I think this paper is well-written, well-analysed with interesting results.

**Other Strengths And Weaknesses:**

n/a

**Questions For Authors:**

1. It is unclear to me how ‘large’ gradient norm could be a proof of better plasticity. I think it’s more persuasive to claim that a ‘stable’ gradient norm rather than the norm itself. If there’s a literature around the connection between gradient norm and plasticity, it’d be nice it they’re included in the Gradient Norm section.
2. On that note, what do you think about the gradient norms consistently increasing in Figure 5? It seems like they will keep increasing when trained beyond 1M steps, and I don’t think that’s something we’d want in terms of stability.
3. Slightly off topic: are there any intuition on why Erdos-Renyi ratio is superior to uniform or other layer-wise ratios? Why is it better to have the ratios linear to the number of input/output neurons rather than quadratic?

**Relation To Broader Scientific Literature:**

Unlocking parameter scaling in RL is a contribution towards a large foundational model in RL.

**Theoretical Claims:**

No theoretical claims were made.

---

> ### Author Rebuttal · Authors · 2025-04-01
>
> Thank you for your positive review. We address your questions below.
>
> > Q1: Related work on iterative pruning
>
> We have already discussed relevant works on dynamic sparse training (DST) in RL within our Introduction and Related Work sections. In the next version, we plan to expand our coverage of more general topology evolution methods, including the papers you mentioned. Thank you for pointing out the valuable references.
>
> > Q2: The connection between gradient norm and plasticity
>
> Your question about distinguishing between 'large' and 'stable' gradient norms is insightful and helps us clarify an important relationship that wasn't fully explained in our manuscript.
>
> In Figure 5, we observe that large dense networks experience a rapid collapse in gradient norm early in training while performance remains poor. This premature descent into a low gradient state reflects the agent's inability to effectively learn from new experiences, directly corresponding to the slow performance improvement in these networks. By contrast, sparse networks maintain more consistent gradient signals throughout training. Therefore, it's this early declining pattern, rather than absolute magnitude, that indicates plasticity loss.
>
> Several studies have explored the relationship between gradient norm collapse and plasticity loss, including:
>
> - Figure 7 in *The Primacy Bias in Deep Reinforcement Learning, ICML 2022*
> - Section 4.2 in *Loss of Plasticity in Continual Deep Reinforcement Learning, CoLLAs 2023*
> - Figure 10 in *Weight Clipping for Deep Continual and Reinforcement Learning, RLC 2024*
> - Figure 5 in *Addressing Loss of Plasticity and Catastrophic Forgetting in Continual Learning, ICLR 2024*
>
> How to precisely measure a network's plasticity loss level remains an open question, which is why current work tends to employ multiple different metrics simultaneously for a more comprehensive assessment. The lower dormant ratios, higher SRank values, and Reset diagnostic experiments in Section 4 collectively demonstrate how sparsity helps preserve network plasticity.
>
> Your question has prompted us to reconsider our presentation. Since gradient norm has confounding relationships with training stability, and our other metrics already establish the plasticity benefits of sparse networks, we plan to move this discussion to the appendix with more detailed explanations in our revised manuscript.
>
> > Q3: Further trend of gradient norms beyond 1M steps
>
> Your concern about continuously increasing gradient norms prompted us to extend our experiments to 2M steps on Dog Trot and Run tasks, shown in [`figure (anonymous link)`](https://anonymous.4open.science/r/ICML_2025_3388/GN.jpg), revealing two interesting patterns:
>
> In Dog Trot, where performance plateaus around 1M steps, gradient norms in sparse networks eventually peak and then gradually decrease, demonstrating natural stabilization.
>
> In Dog Run, where learning continues beyond 1M steps, gradient norms maintain moderate activity, corresponding directly with ongoing performance improvements.
>
> These patterns suggest sparse networks achieve an effective balance: they maintain sufficient gradient activity for learning without collapsing (unlike dense networks), while also not becoming unstable over time. We'll include this extended analysis in our revised manuscript's appendix to better illustrate the long-term benefits of network sparsity.
>
> > Q4: Comparison between Erdos-Renyi ratio and other layer-wise ratios
>
> Based on sparse training research in the broader deep learning field, the advantage of ER initialization likely stems from providing more balanced information flow across network layers. By scaling sparsity proportionally to the geometric mean of input/output dimensions (rather than uniformly), ER maintains approximately equal fan-in/fan-out ratios across layers, preventing bottlenecks in both forward and backward passes.
>
> The core purpose of our paper is to demonstrate how network sparsity as a fundamental property helps scale up DRL model size, rather than comparing different sparsification methods. We chose ER initialization because previous sparse training studies (in both DRL and supervised learning) have established its superiority over uniform initialization. To directly compare these approaches in our setting, we conducted additional experiments shown in the [`figure (anonymous link)`](https://anonymous.4open.science/r/ICML_2025_3388/ER_Uniform_Ratios.jpg).
>
> The results reveal that at lower sparsity levels (≤0.6), both initialization methods perform comparably. However, at higher sparsities (≥0.8), uniform layer-wise ratios exhibit a dramatic performance drop. These findings suggest that:
>
> 1. Sparsity itself benefits DRL scalability regardless of the specific layer-wise ratio configuration.
> 2. ER initialization provides greater robustness, especially at high sparsity levels, by creating a more balanced network topology that better maintains information flow.

---

### Official Review · Reviewer_BkgJ · 2025-03-13

**Overall Recommendation:** 4

**Summary:**

This paper explores the scalability benefits of incorporating static network sparsity in deep reinforcement learning models. It introduces one-shot random pruning, where a fixed proportion of network weights are removed before training, leading to improved parameter efficiency compared to scaling up dense architectures. The analysis highlights that sparse networks enhance expressivity while alleviating optimization challenges such as plasticity loss and gradient interference. Furthermore, experiments on visual and streaming RL tasks demonstrate the robustness of sparsity, showcasing its consistent advantages across diverse reinforcement learning scenarios.

## update after rebuttal
The authors provided additional results of DER on Atari, which also shows the sign of the proposed findings. I'll keep my positive evaluation.

**Claims And Evidence:**

I'm not familiar with the analysis in Sections 4.3 and 4.4, but overall, I believe the corresponding experiments effectively support the claims presented in each section.

**Essential References Not Discussed:**

No

**Experimental Designs Or Analyses:**

I appreciate that Appendix B provides detailed settings for the major experiments. While using eight random seeds per experiment is slightly lower than expected, it is still a reasonable choice.

**Methods And Evaluation Criteria:**

The primary experiments are conducted on the four most challenging DMC tasks using two RL algorithms, SAC and DDPG. Additionally, the authors extend their evaluation to visual and streaming RL. Overall, the evaluation appears solid and well-founded.

**Other Comments Or Suggestions:**

I appreciate the authors' detailed analysis of RL methods for continuous action control problems. It would be interesting to see these observations extended to value-based methods in discrete action spaces, such as DQN on Atari. Nonetheless, the current findings are valuable and worth sharing with the community.

**Other Strengths And Weaknesses:**

N/A

**Questions For Authors:**

No

**Relation To Broader Scientific Literature:**

This paper presents insightful and valuable findings on scaling deep reinforcement learning networks. It demonstrates that incorporating static network sparsity through simple one-shot random pruning can enhance scalability and outperform dense counterparts. These findings are not only relevant to DRL but may also have broader implications for other learning paradigms, such as self-supervised learning and language model post-training.

**Theoretical Claims:**

N/A

---

> ### Author Rebuttal · Authors · 2025-04-01
>
> Thank you for your thorough review and positive evaluation of our work.
>
> > It would be interesting to see these observations extended to value-based methods in discrete action spaces, such as DQN on Atari. Nonetheless, the current findings are valuable and worth sharing with the community.
>
> We agree that extending our findings to value-based methods with discrete action spaces is crucial for demonstrating the broader applicability of network sparsity benefits. To address this, we've conducted new experiments on the Atari-100k benchmark using Data Efficient Rainbow DQN (DER) [1] as our baseline algorithm.
>
> The experiments compare performance improvements when scaling network width to 3x the default size with varying sparsity levels (0.0, 0.4, and 0.8). Results are shown in [`figure (anonymous link)`](https://anonymous.4open.science/r/ICML_2025_3388/atari_sparsity_results.pdf). *(Note: Due to time constraints before the first round response deadline, we've only completed experiments on 13/26 Atari tasks. We expect to update the figure with complete results within one more day.)*
>
> These results show that introducing network sparsity while scaling up model size produces similar benefits in discrete action tasks as observed in our continuous control experiments. This further validates the general effectiveness of network sparsity for scaling DRL models across different domains and algorithms.
>
> We note that the Atari-100k low-data regime may not fully demonstrate the benefits of scaling, and more comprehensive studies with longer training (e.g., 10M environment steps) would be valuable for future work [2]. Nevertheless, these preliminary results provide additional evidence supporting our main findings about network sparsity's role in unlocking DRL scaling potential.
>
> [1] When to use parametric models in reinforcement learning?, NeurIPS 2019
>
> [2] In value-based deep reinforcement learning, a pruned network is a good network, ICML 2024

---

> > ### Comment · Reviewer_BkgJ · 2025-04-04
> >
> > Thank you for the additional experimental results of DER. I appreciate your effort and will maintain my positive evaluation. Great work!

---

### Official Review · Reviewer_W4Bt · 2025-03-13

**Overall Recommendation:** 4

**Summary:**

This paper uncovers an interesting finding: Instead of pursuing more complex modifications, introducing static Network sparsity alone can unlock further scaling potential beyond their dense counterparts with state-of-the-art architectures.  And in experiments, they show that only using one-step random pruning can achieve great performance in several commonly used benchmarks.

**Claims And Evidence:**

Yes

**Essential References Not Discussed:**

I think the related works analyzed in this paper are wide.

**Experimental Designs Or Analyses:**

I checked the experimental designs, and I think they are valid.

**Methods And Evaluation Criteria:**

Their Method makes sense for the problem.

**Other Comments Or Suggestions:**

One thing I did not find in the paper is which part of the AC architecture should be more important -- actor or critic. What would happen to the learning ability when you only prune the critic or actor? Does a sparse actor play a key role?

**Other Strengths And Weaknesses:**

1) Well-written, clear summary
2) completely experimental analysis
3) rich and solid visualization metrics testing
4) makes an important contribution to the understanding of the scaling law in Deep RL community

**Questions For Authors:**

Please see above

**Relation To Broader Scientific Literature:**

I think this paper made a general contribution to the scaling law of the AGI.

**Theoretical Claims:**

The theory in the paper is solid.

---

> ### Author Rebuttal · Authors · 2025-04-01
>
> Thank you very much for your positive evaluation of our work.
>
> > Q: What would happen to the learning ability when you only prune the critic or actor? Does a sparse actor play a key role?
>
> This is an excellent question about the relative importance of sparsity in different components of actor-critic architectures. We conducted additional experiments comparing four configurations: dense networks, sparse actor only, sparse critic only, and sparse for both actor and critic.
>
> Our results shown in [`figure (anonymous link)`](https://anonymous.4open.science/r/ICML_2025_3388/Actor%20vs%20Critic.jpg) clearly show that applying sparsity to both actor and critic networks yields substantially better performance than either partial approach. Interestingly, when sparsity is applied to only one component (either actor or critic), the scaling curves closely resemble those of the dense baseline.
>
> This suggests that scaling benefits emerge from a balanced application of sparsity across the entire architecture rather than from any single component. We believe this occurs for several reasons:
>
> 1. Critics require sparsity to avoid optimization pathologies during online TD learning. As demonstrated in Section 4, scaling up dense DRL models leads to severe plasticity loss in critics and diminished value representation capacity.
> 2. The need for sparse actors aligns with previous findings [1,2] showing that actors are particularly sensitive to network scaling. In actor-critic methods, since actor learning depends on critic outputs, unnecessarily complex actor networks can impede performance. The specific mechanisms behind sparse actor benefits warrant further investigation.
> 3. The balance between actor and critic parameters is crucial. Our experiments use the SimBa architecture, which carefully established optimal component sizing [1] - in the default 4.51M parameter configuration, the critic accounts for 4.34M parameters. When scaling our networks, we proportionally increased both actor and critic sizes, already accounting for their different representational capacity requirements. By maintaining this ratio during scaling, applying sparsity to both components becomes necessary to preserve the architectural balance that makes SimBa effective.
>
> [1] SimBa: Simplicity Bias for Scaling Up Parameters in Deep Reinforcement Learning, ICLR 2025
>
> [2] Bigger, Regularized, Optimistic: scaling for compute and sample-efficient continuous control, NeurIPS 2024

---

### Official Review · Reviewer_yHy2 · 2025-03-17

**Overall Recommendation:** 5

**Summary:**

This paper shows that current deep RL architectures result in performance decreases when scaling the network in width or depth. Introducing sparsity in the form of a fixed mask over the weights, is able to to resolve this issue. The source of the benefits of sparsity is investigated in terms of representational capacity, plasticity and gradient interference. O
Further experiments demonstrate the utility of the method in visual RL and streaming RL.

**Claims And Evidence:**

This paper has very comprehensive experiments and does an excellent job at demonstrating the usefulness of introducing sparsity.
The variety of ablations and analyses help support the main points and also give some nice insights into why this method may be working.

**Essential References Not Discussed:**

None

**Experimental Designs Or Analyses:**

Extensive analyses are done on various aspects of the method (sparsity ratio, effect of scale) with appropriate experimental choices.
Deeper looks into the effect on metrics such as reprentational capacity, plasticity (dormant neurons, gradient norms), gradient interference and simplicity bias give a more comprehensive view of the effects of sparsity.

**Methods And Evaluation Criteria:**

The benchmarks chosen are standard and appropriate for this setting.

**Other Comments Or Suggestions:**

None

**Other Strengths And Weaknesses:**

The presentation is very clear and visually-pleasing, with nice use of "callout blocks" of different colors to highlight certain takeaways or important points.

The writing and organization are also excellent and the paper is information-dense but still easy to follow.

A weakness of the paper is that most of the experiments are conducted on four environments from DMC, which may impact the generalizability of the analyses. There are other experiments with visual and streaming RL though, which can support the idea that sparsity would still be useful in other settings. Overall, I think the currente experiments are adequate.

**Questions For Authors:**

Some clarification questions:
- Did you try the more naive appraoch of a fixed sparsity ratio for every layer rather than the Erdos-Renyi ratio? Is this ineffective or only less effective?

- In the Lottery Ticket Hypothesis paper, the authors do not observe any performance benefits when introducing sparsity with random weights (rather than the winning ticket). How would you reconcile their result with this paper's results, where large benefits seem to be observed?

**Relation To Broader Scientific Literature:**

This paper makes a surprising observation: static sparsity can result in large performance gains and scaling.
This finding could make RL researchers rethink how to design new network architectures and I think these results would be of great interest to the RL community.

**Theoretical Claims:**

N/A

---

> ### Author Rebuttal · Authors · 2025-04-01
>
> > **Q1**: Limited DMC environments affecting generalizability
>
> We acknowledge the reviewer's concern that most experiments are conducted on DMC, which might impact the generalizability of our analyses. To address this, we've conducted new experiments on the Atari-100k benchmark using Data Efficient Rainbow DQN (DER) [1] as our baseline algorithm.
>
> The experiments compare performance improvements when scaling network width to 3x the default size with varying sparsity levels (0.0, 0.4, and 0.8). Results are shown in [`figure (anonymous link)`](https://anonymous.4open.science/r/ICML_2025_3388/atari_sparsity_results.pdf).
>
> These results show that introducing network sparsity while scaling up model size produces similar benefits in discrete action tasks as observed in our continuous control experiments. This further validates the general effectiveness of network sparsity for scaling DRL models across different domains and algorithms.
>
> We note that the Atari-100k low-data regime may not fully demonstrate the benefits of scaling, and more comprehensive studies with longer training would be valuable for future work. Nevertheless, these preliminary results provide additional evidence supporting our main findings about network sparsity's role in unlocking DRL scaling potential.
>
> [1] When to use parametric models in reinforcement learning?, NeurIPS 2019
>
> > **Q2**: Fixed uniform sparsity ratios vs. Erdos-Renyi ratios
>
> Based on the reviewer's question, we conducted new experiments comparing the naive approach of fixed/uniform sparsity ratios across all layers with the Erdos-Renyi (ER) ratio. The results shown in the [`figure (anonymous link)`](https://anonymous.4open.science/r/ICML_2025_3388/ER_Uniform_Ratios.jpg) reveal that at lower sparsity levels (≤0.6), both initialization methods perform comparably, while at higher sparsities (≥0.8), uniform layer-wise ratios exhibit a dramatic performance drop.
>
> The advantage of ER initialization stems from its ability to provide more balanced information flow across network layers. By scaling sparsity proportionally to the geometric mean of input/output dimensions, ER maintains approximately equal fan-in/fan-out ratios across layers, preventing bottlenecks that emerge with uniform sparsity at high sparsity levels where network connectivity becomes critical for information propagation.
>
> We want to emphasize that our paper's focus is on demonstrating how network sparsity as a fundamental property enables DRL model scaling, rather than identifying the optimal sparse topology. The fact that uniform sparsity at appropriate levels also improves performance further supports our main claim that sparsity itself (regardless of specific implementation) is a key enabler for unlocking the scaling potential of DRL networks.
>
> > **Q3**: Random sparsity benefits vs. Lottery Ticket Hypothesis findings
>
> The reviewer raises an important question about reconciling our findings with the Lottery Ticket Hypothesis (LTH). The key difference lies in the fundamentally different problem settings and objectives:
>
> First, we must distinguish between previous sparse training studies and our work. Earlier studies on sparse training (including LTH) were motivated by model compression to reduce computational costs while maintaining performance. Such approaches operate under the assumption that in supervised or unsupervised learning, larger dense networks generally yield better performance—the core premise behind modern scaling laws.
>
> However, this assumption is violated in online DRL settings. As demonstrated in Figure 1 of our manuscript, online DRL networks face severe scaling barriers where increasing model size not only fails to improve performance but often leads to catastrophic collapse. Our analysis in Section 4 further reveals that these scaling barriers emerge because larger dense networks are more susceptible to optimization pathologies during online RL training, preventing them from leveraging their theoretical capacity.
>
> Our work aims to demonstrate that network sparsity itself, as a fundamental network property, can mitigate these pathologies and unlock the scaling potential of DRL networks. This represents a fundamentally different objective than LTH's focus on finding efficient subnetworks within larger models.
>
> Therefore, our findings don't contradict the Lottery Ticket Hypothesis but rather highlight the unique utility of network sparsity in addressing online DRL's specific pathologies and inscalability. For a given sparsity level, we believe that better topologies (such as "winning tickets") could outperform one-shot random pruning. However, finding tickets requires significant computational resources, and discovering "lottery tickets" without training remains challenging. Efficiently identifying winning tickets as trainable networks represents valuable future work, but identifying optimal sparse topologies was not the focus of our current study.

---

### Decision · Program_Chairs · 2025-05-01

**Decision:**

Accept (oral)

**Comment:**

This paper shows that scaling sparse neural networks trained with static network sparsity not only improves parameter efficiency but also enhances resistance to plasticity loss and gradient interference, compared to scaling dense neural networks. All reviews are very positive, recognize the novelty of the findings, and appreciate the presentation of the paper. Therefore, the AC suggests acceptance and encourages the authors to include the rebuttal discussion in the final version of the paper.